# Sarcoconvolutums F and G: Polyoxygenated Cembrane-Type Diterpenoids from *Sarcophyton convolutum*, a Red Sea Soft Coral

**DOI:** 10.3390/molecules27185835

**Published:** 2022-09-08

**Authors:** Tarik A. Mohamed, Abdelsamed I. Elshamy, Mohamed H. Abd El-Razek, Asmaa M. Abdel-Tawab, Sherin K. Ali, Mohamed Aboelmagd, Midori Suenaga, Paul W. Pare, Akemi Umeyama, Mohamed-Elamir F. Hegazy

**Affiliations:** 1Chemistry of Medicinal Plants Department, National Research Centre, 33 El-Bohouth St., Dokki, Giza 12622, Egypt or; 2Department of Natural Compounds Chemistry, National Research Centre, 33 El-Bohouth St., Dokki, Giza 12622, Egypt; 3Marine Biotechnology and Natural Products Laboratory, National Institute of Oceanography and Fisheries, Cairo 11516, Egypt; 4Pharmacognosy Department, National Research Centre, 33 El-Bohouth St., Dokki, Giza 12622, Egypt; 5Faculty of Pharmaceutical Sciences, Tokushima Bunri University, Yamashiro-cho, Tokushima 770-8514, Japan; 6Department of Chemistry & Biochemistry, Texas Tech University, Lubbock, TX 79409, USA

**Keywords:** soft coral, *Sarcophyton convolutum*, diterpenenoid, cembranoids, cytotoxic activity

## Abstract

Natural products and chemical analogues are widely used in drug discovery, notably in cancer and infectious disease pharmacotherapy. *Sarcophyton convolutum* (Alcyoniidae) a Red Sea–derived soft coral has been shown to be a rich source of macrocyclic diterpenes and cyclized derivatives. Two previously undescribed polyoxygenated cembrane-type diterpenoids, sarcoconvolutums F (**1**) and G (**2**), as well as four identified analogues (**3**–**6**) together with a furan derivate (**7**) were isolated from a solvent extract. Compounds were identified by spectroscopic techniques, including NMR, HREIMS, and CD, together with close spectral comparisons of previously published data. Sarcoconvolutum F (**1**) contains a rare 1-peroxid-15-hydroxy-10-ene functionality. Isolated metabolites (**1**–**7**) were screened against lung adenocarcinoma (A549), cervical cancer (HeLa) and oral cavity carcinoma (HSC-2) lines. Compound **4** exhibited an IC_50_ 56 µM and 55 µM against A549 and HSC-2 cells, respectively.

## 1. Introduction

Drug development relies heavily on natural products and structural analogues, particularly for cancer and infectious disease pharmacotherapy’s [1]. While terrestrial plants and soil microorganisms are traditionally a rich source of biologically-active natural products, marine organisms also produce a broad assortment of chemical defenses with significant chemical diversity and activity [2]. The Red Sea with both tropical and subtropical ecosystems supports robust and diverse marine life including rich native biota. Indeed, while over 180 soft coral species have been discovered around the world, a significant percentage reside and are native to the Red Sea [3]. Soft coral are traditionally catagorized based on a sclerite classification system; these sclerites are small aggregates of calcium carbonate embedded in the soft coral tissue that provide colony support. Having eight tentacles, they are classified in the class Octocorallia. Alcyonacean, an order of Octocorallia are massive sessile invertebrates with an unique stalk and a mushroom-shaped capitulum [4,5]. The genus *Sarcophyton,* in the order Alcyonacea includes 35 species, six of which were recently recognized as new [6,7,8,9,10,11]. The chemistry and pharmacology of *Sarcophyton* metabolites have been investigated extensively. Sesquiterpenes, diterpenes, steroids and fatty acids have been proposed to be a major source of pharmacological activity [12,13,14]. Indeed, macrocyclic cembranes and their derivatives, have been confirmed to be potent anticancer, antibacterial, anti-inflammatory, anti-osteoporotic, antimetastatic, antiangiogenic, and neuroprotective natural bioactive agents [15]. Diterpenoids from the cembrane family serve in reef chemical defense against predators or competitors as they are often highly cytotoxic [13,15,16]. The genus includes 39 biscembranoids, 323 diterpenes, 11 sesquiterpenes, 53 polyoxygenated sterols, and 55 miscellaneous metabolites, many of which have documented pharmacological efficacies [17].

Cembrane-type diterpenoid biosynthesis utilizes a geranylgeraniol precursor that is cyclized between carbons 1 and 14, generating a 14-membered cembrane or thumbergane skeleton [18]. An *E* double bond geometry from the geranylgeraniol substrate generates a (+)-cembrene, that was first isolated from pine oleoresin [19]. Kobayashi and colleagues later isolated the cembranoid diterpene, sarcophytol A, a potent antitumor agent from Okinawan soft coral *S. glaucum* [20]. Subsequently, hundreds of cembranoids have been isolated and identified from insects, and marine organisms alike [15]. Cembranoids generally include a cyclic ether and a lactone, or furane moiety usually present between C-1, C-2, C-15 and C-16 (type I) or C-2, C-3, C-4 and C-18 (type II). At C-17, the α-position of an α,β-unsaturated butenolactone moiety, is frequently substituted with a Me group or an *exo*-CH_2_= group in type-I cembranoids. A CH_2_OH group can exist as a linkage intermediate between a Me and a terminal C=C bond. Several cembranoids have been reported as type II. Sarcophyocrassolide A, isolated from a CHCl_3_ extract of *S. crassocaule*, was the first cembranolide skeleton bearing an 8-hydroperoxy group in the α-methylidene-γ-lactone skeleton [21]. 

In this study, the soft coral *S. convolutum,* native to and collected from the Red Sea was solvent extracted and chemically characterized. A series of cembrene diterpenoids were identified (Figure 1) and metabolites were assayed for cytotoxicity against a series of tumor cell lines. 

## 2. Results

### Chemical Elucidation

A soft coral solvent extract was chromatographically separated, yielding cembrene diterpenoids, sarcoconvolutums F (**1**) and G (**2**), as well as recognized compounds; sarcophine (**3**) [14,22], 7α,8β-dihydroxydeepoxysarcophine (**4**) [23], crassumol G (**5**) [24], sarcophyolide E (**6**) [25], (*S*)-5-hydroxy-3,4-dimethyl-5-pentylfuran-2(5*H*)-one (**7**) [26]. Spectroscopy analysis identified the chemical configuration (Figure 1). The carbon skeleton was similar, according to NMR data, with the degree of oxidation and the configuration of one or more chiral centers differing. The assumption of a cembranoid-skeleton backbone was based on precedent from soft coral literature.

Compound **1** was generated as a colorless oil with a positive optical rotation in methanol **${\left[\alpha \right]}_{D}^{25}$** + 41.0 in MeOH. Based on HREIMS data that exhibited an [M]^+^ ion at *m*/*z* 400.2094 (calcd *m*/*z* 400.2098), the chemical formula was estimated to be C_20_H_32_O_8_, reflecting a five degrees of unsaturation. Two double bonds and one carbonyl from NMR data are responsible for three of the five elements of unsaturation, giving rise in a bicyclic molecule. An absorption for a hydroxyl (υ_max_ 3450 cm^−1^), a carbonyl (υ_max_ 1750 cm^−1^) and an olefin (υ_max_ 1669 cm^−1^) were confirmed by IR analysis. The ^1^H NMR spectrum (Table 1) exhibited two oxygenated protons at δ_H_ 5.04 (d; *J* = 10.4 Hz); and δ_H_ 3.64 (d; *J* = 10.9 Hz). Three olefinic protons at δ_H_ 5.64 (dd; *J* = 15.7, 3.7 Hz), 5.61 (br d; *J* = 15.7 Hz) and 5.22 (br d; *J* = 10.4 Hz); four signals at δ_H_ 1.80 s, 1.69 s, 1.21 s, and 1.16 s were identified as methyls. Twenty carbon signals were observed in the ^13^C NMR spectrum (Table 1), which were further differentiated by DEPT to four methyls (three oxygenated at δ_C_ 22.7, 22.9, 27.3 and one olefinic at δ_C_15.9), five methylenes (19.0, 24.9, 26.7, 36.1, and 43.3), five methines (two oxygenated at δ_C_ 69.7 and 80.9, three olefinic at δ_C_ 117.5, 126.6 and 136.3), and six quaternary carbons (four oxygenated at δ_C_ 73.9, 77.5, 79.9 and 86.9, one olefinic at δ_C_ 143.5 and one keto at δ_C_175.2). The cembrene-based diterpenoid was generated using the 1D NMR analysis described above of **1** [14,23,27,28,29,30]. 

Combining spectral data of compound **1** indicated to a cembranoid diterpene molecular framework containing a rare 1-peroxid-15-hydroxy-10-ene with a close similarity to sarcoroseolide B previously isolated from *S. roseum* [31], with the exception that **1** was missing the oxygen bridge which was found in sarcoroseolide B expected to be replaced by the peroxy group at C-1. Moreover, compound **1** possessed an 34 amu in its molecular weight increases than sarcoroseolide B, which predict that **1** had a peroxy group. Besides, functionalities with a molecular formula of sarcoroseolide B suggesting a tricyclic structure compared with **1** validated by HREIMS to be a bicyclic frame skelton. Spectral study of ^1^H- ^1^H COSY and HMBC validated the predicted structure. The signal at δ_H_ 5.04 (d, *J* = 10.4 Hz, 1H) correlated with a proton signal at δ_H_ 5.22 (d, *J* = 10.4 Hz, 1H) and quaternary olephnic carbon at δ_C_ 143.5 (Figure 2), respectively, allowed for the assignments of H-2, H-3, C-4 [14,27,29]. The long-range HMBC correlations (Figure 2) between H_3_-17 and carbon signals at δ_C_ 86.9 (C-1), δ_C_ 77.5 (C-15) and keto group at δ_C_ 175.2 (C-16) indicated the missing double bond resulted in a saturated lactone ring and consequently led the oxygenated tertiary carbons (δ_C_ 86.9 and 77.5) at C-1 and C-15 [32]. HMBC correlation of the methyl protons H-19 (δ_H_ 1.21) correlated with C-7 (69.7)/C-8 (73.9)/C-9 (43.3), and H-9 (δ_H_ 2.12) and C-7 (69.7) confirming placement of the hydroxyl groups carbons at C-7 and C-8. HMBC correlations (Figure 2) were also observed between methyl protons H-20 (δ_H_ 1.16 s) and C-12 (79.9), C-11 (136.3), C-13 (26.7), H-13 (δ_H_ 1.92) and C-11 (136.3) and the olefinic proton signal at δ_H_ 5.61 (H-11, br d, *J* = 15.7 Hz) showed an HMBC correlation with olefinic carbon at δ_C_ 126.6 (C-10) and H-10 (δ_H_ 5.64) and C-9 (43.3) establishing that the hydroxyl and the double bond functionalities are located at C-12 and C-10/C-11, respectively. The suggested relative configurations of **1** were determined by NOESY experiments and based on coupling constants (Figure 3). The vicinal coupling constant of 10.4 Hz between H-3 (δ_H_ 5.22) and H-2 (δ_H_ 5.04) as well as a NOESY correlation of H-2/H-3 (δ_H_ 5.22) established a *cis* configuration between the γ-lactone ring and the vinylic proton (H-3). As no correlation was observed between H-2α and the teriary methyl (Me-17), improved that both of H-2α and Me-17 are in an *anti*-orientation. A cross peak correlation between H_3_-19 and H_3_-20, along with comparison with the stereochemistry of sarcoroseolide B established that, both of them are in β-orientation. 1*S**, 2*R**, 7*R**, 8*R**, 12*S**, and 15*R** the relative configurations of **1** were established based on the aforementioned observations and extensive assessment of other NOESY interactions. CD spectrum comparison of **1** and sarcophine (**3**) revealed the opposite absolute configuration for the two compounds at C-2, corroborating CD findings for **1** (Figure 4), [14,23]. From the above spectral data, **1** was established as sarcoconvolutum F.

A combination of the intensity of the ^1^H and ^13^C NMR signals for **2**, together with the [M]^+^ ion at *m*/*z* = 332.1991 (calcd. 332.2222) in the HREI mass spectrum, supported a chemical formula of C_20_H_28_O_4_ corresponding to seven double bond equivalents. Compound **2** was produced as a colorless oil with a negative optical rotation of **${\left[\alpha \right]}_{D}^{25}$** − 3.4 in methanol. The IR spectra revealed two distinct bands at 3450 cm^−1^ (OH) and 1750 cm^−1^ (CO). The ^13^C NMR spectrum displayed twenty carbon signals (Table 1), which were further differentiated by DEPT to three methyls (one oxygenated at δ_C_ 19.8 and two olephenic at δ_C_ 8.8, 20.7), seven methylenes (one exomethelene at δ_C_ 111.1 and six aliphatic at δ_C_ 26.8, 27.2, 30.8, 32.9, 33.4 and 36.1), four methines (three oxygenated at δ_C_ 74.6, 79.5 and 83.1, one olefinic at δ_C_ 120.4), and six quaternary carbons (one oxygenated at δ_C_ 85.1, one keto at δ_C_ 174.8, four olefinic at δ_C_ 125.0, 145.9,148.6 and 162.0). The presence of an ether linkage was suggested by a high downfield carbon signals at δ_C_ 83.1 and 85.1, which were functionally validated by HREIMS [24]. Seven degrees of unsaturation were deduced, suggesting a tricyclic cembrane frame skeleton. 

The assignment of H-2, H-3, C-4, and C-1 of a cembrene diterpenoid was made possible by the correlation of the olefinic signal at δ_H_ 5.09 (d, *J* = 9.8 Hz) with the oxygenated proton at δ_H_ 5.09 (d; *J* = 9.8 Hz) and quaternary olefinic carbons at δ_C_ 145.9 and 162.0 in ^1^H-^1^H COSY. The HMBC linkage of a methyl signal at δ_H_ 1.85 (s) with C-1 and a keto group at δ_C_ 174.8 suggested the presence of a lactone ring comprising C-1/C-2, allowing H-17 and C-16 to be assigned. The HMBC correlation of the olefinic methyl group at δ_H_ 1.96 (H-18) with an olefinic methine at δ_C_ 120.4 (C-3) and a methylene signal at δ_C_ 32.9 allowed for the assignment of H-18 (δ_H_ 1.96, s) and H-5 [δ_H_ 2.08 (1H, d, *J* = 10.4 Hz) and 2.43 (1H, d, *J* = 14.1 Hz), which was validated by HMQC analysis. The HMBC correlation of an oxygenated methyl group at δ_H_ 1.16 (H-19) with oxygenated methine at δ_C_ 74.6 (C-7), oxygenated quaternary carbon at δ_C_ 85.1 (C-8) and aliphatic methylene at δ_C_ 36.1 (C-9) and the data comparison with those of crassumol G [24], revealed that the sole difference between **2** and crassumol G was in the stereochemistry. On the basis of coupling constants and NOESY experiments, the relative configuration of **2** was modeled. A *cis* configuration between the γ-lactone (H-2) and the olefinic proton (H-3) was established using a vicinal coupling constant of 9.8 Hz between H-2 (δ_H_ 5.47) and H-3 (δ_H_ 5.09) as well as a NOESY correlation of α-orientation of H-2 with H_3_-18 (δ_H_ 1.96). 

The value of vicinal coupling constant which equally to 10.1 Hz between H-7 (δ_H_ 3.22) and H-6 (δ_H_ 1.58), as well as 13.7 Hz vicinal coupling between H-6 (δ_H_ 1.99) and H-5 (δ_H_ 2.08) with the aid of NOESY spectrum supported the orientation of H-7 in α-orientation and differentiate between C-6 protons chemical shifts together with the comparison with those of crassumol G [24]. The NOESY correlations observed between H-14 (δ_H_ 2.32)/H-2, H_3_-17 as well as a NOESY correlation H-14 (δ_H_ 2.66)/H-20a (δ_H_ 4.88), and H-20b (δ_H_ 5.04)/H-11(δ_H_ 4.50) and H-9 (δ_H_ 1.72)/H_3_-19, H-7 (δ_H_ 3.22) showed that H-2, H-7, H-11, H_3_-17 and H_3_-19 were in the α-orientation. Thus, the isolate **2** has an 2*R**, 7*S**, 8*S**, and 11*R** relative configurations based on the aforementioned observations and extensive assessment of other NOESY interactions. Supporting CD data for **2** (Figure 4) comparisons between **1**, **2** and sarcophine (**3**); compound (**2**) indicated the same absolute configuration for **1** and reverse absolute configuration for sarcophine **3** at C-2 [14,23]. From the above spectral data, **2** was established as sarcoconvolutum G.

Despite the fact that many cembrane-type diterpenoids have been identified from soft corals in the genus *Sarcophyton*, the solvent extract of the soft coral *S. convolutum* resulted in the isolation of two novel cembrane diterpenoids (**1**–**2**), one of which, sarcoconvolutum F (**1**), has a rare 1-peroxid-15-hydroxy-10-ene cembrane skeleton. Putative biosynthetic pathways are postulated as depicted in Figure 5, to explain the biogenetic origin of the secondary metabolites (**2**–**6**), along with **1** which is outlined separately in Figure 6. An alternative biosynthetic route would be that the enolate is oxidized directly to form an epoxide, leading to an alpha-hydroxy ketone via an epoxide intermediate (not shown). Besides the previously described phytochemistry from the same species (Sarconvolutum A, B, D & E) [1], the metabolites may all be traced back to a common precursor, sarcophine (**3**) which is a typical cembrane diterpenoids discovered in many soft coral belonging to the Sarcophyton genus and accumulating from various locations [25]. Sarcophine is a common compound for this soft coral species and can be used as a chemotaxonomic tool. We propose a biogenetic pathway to **1** that begins with the nucleophilic attack of α,β-unsaturation-γ-lactone moeity to form tetra-substituted epoxide, followed by the production of a hemiketal intermediate to convert the double bond to 1-peroxy-2-hydroxy, a rare constituent of sarcoconvolutum F (**1**).

Three cancer cell lines (A549, HeLa, and HSC-2) were used to investigate the cytotoxicity of **1**–**5** at three doses (100, 10, and 1 µM). Compound **4** was cytotoxic to cell lines A549 and HSC-2. At dosages ranging from 0.001 to 100 µM, the A549 and HSC-2 cell lines were examined for dose-dependent toxicity. In A549 and HSC-2 cells, **4** yielded IC_50_ values of 56 μM and 55 μM, respectively.

## 3. Materials and Methods

### 3.1. General Procedures

A JASCO P-2300 polarimeter was used to measureoptical rotation (Jasco Corporation, Tokyo, Japan). A Shimazi FTIR-8400S was used to measure IR spectra (Columbia, MD 21046, USA). A Bruker 600 or 500 Hz NMR spectrometer was used to record 1D and 2D NMR spectra (MA, USA). Chemical shifts were expressed in parts per million (ppm), while coupling constants were expressed in hertz (Hz). On a JEOL JMS-700 instrument, HR-MS spectra were collected (Tokyo, Japan). A JASCO 810 polarimeter was used to measure electronic circular dichroism (ECD). For column chromatography, Merck’s Silica Gel 60 (230–400 mesh, Merck, Darmstadt, Germany) was employed. TLC examination was performed on precoated silica gel plates (Merck, Kieselgel 60 F254, 0.25 mm, Merck, Darmstadt, Germany). A Jasco PU-980 pump clever HPLC pump was used to perform high-performance liquid chromatography (HPLC).

### 3.2. Animal Material

In March 2017, the soft coral *S. convolutum* was collected from the Red Sea coast in Hurghada, Egypt. A voucher specimen (08RS1071) from the National Institute of Oceanography and Fisheries’ marine biological station in Hurghada, Egypt, was the basis for coral identification (by M Al-Hammady).

### 3.3. Extraction and Isolation

*Sarcophyton conovlutum* (1.5 kg, total wet weight) was rapidly frozen in a −20 °C chamber and kept frozen until extraction. Tissue was extracted with ethyl acetate at room temperature (3 L × 3 times) and then subjected to gravity chromatography on an octadecylsilyl (ODS) column (6 × 120 cm). The solvent gradient started with 100% water and shifted by 10% increments with methanol until 100% MeOH with a total of 11 fractions generated. Fractions were monitored by TLC and combined to give 10 main fractions as follows: SAC-10 (0.75 g), SAC-20 (2.1 g), SAC-30 (1.9 g), SAC-40 (2.1 g), SAC-50 (2.5 g), SAC-60 (3.2 g), SAC-70 (1.1 g), SAC-80 (1.0 g), SAC-90 (2.4 g), and SAC-100 (1.9 g). Fraction SAC-40 (2.1 g) was HPLC purified in MeOH/H_2_O (35:65 *v*/*v*).The flow rate was set to 1.5 mL/min and was at 0–70 min to afford 1 (10 mg, purity > 99% by HPLC), (eluent hexane/EtOAc 2:1, Rf = 0.35), 2 (9 mg purity > 95% by HPLC), (eluent hexane/EtOAc2:1, Rf = 0.45), 3 (7.5 mg purity > 96% by HPLC), (eluent hexane/EtOAc2:1, Rf = 0.55), 4 (7 mg purity > 92% by HPLC), (eluent hexane/EtOAc 2:1, Rf = 0.52), 5 (11 mg purity > 95% by HPLC), (eluent hexane/EtOAc 2:1, Rf = 0.43), 6 (7 mg purity > 94% by HPLC), (eluent hexane/EtOAc2:1, Rf = 0.56), 7(14 mg purity > 98% by HPLC), (eluent hexane/EtOAc 2:1, Rf = 0.55).

### 3.4. Spectroscopic Data

#### 3.4.1. Sarcoconvolutum F(**1**)

Colorless oil; **${\left[\alpha \right]}_{D}^{25}\;$** + 41.0 (c 1, MeOH); IR (KBr) ν_max_ 3351, 3000, 1745, 1710, 1685, 1670, 1255 cm^−1^; 1H NMR and 13C NMR data, see Table 1; HREIMS m/z 400.2094 [M ]^+^ (calcd. 400.2121, C_20_H_32_O_8_).

#### 3.4.2. Sarcoconvolutum G (**2**)

Colourless oil; **${\left[\alpha \right]}_{D}^{25}\;$** − 3.4 (c 1, MeOH); IR (KBr) ν_max_: 3450, 2931, 1746, 1453, and 1233 cm^−1^; 1H and 13C NMR data, see Table 1; HREIMS *m*/*z* 332.1991 [M]^+^ (calcd. 332.2222, C_20_H_28_O_4_).

### 3.5. Cell Culture and Treatment Conditions

Squamous cell carcinoma of the oral cavity (HSC-2), non-small cell lung adenocarcinoma (A549), and human cervical cancer cell (HeLa) (ATCC^®^) were grown as monolayers in Dulbecco’s modified Eagle’s medium (DMEM) supplemented with 10% FBS, 4 mM l-glutamine, 100 U/mL penicillin, and 100 g/mL streptomycin sulphate Monolayers at 70–90 percent confluence were passed using a trypsin-EDTA solution. In a humidified CO_2_ incubator with 5% CO_2_, cell were held at 37 °C. All materials and reagents for the cell cultures were purchased from Lonza (Verviers, Belgium).

#### 3.5.1. Cytotoxicity Assay

A modified MTT (3-[4,5,[4,5-dimethylthiazole-2-yl]-2,5-diphenyltetrazolium bromide) test based on a previously published technique [33] was used to assess forcytotoxicity. A96-well plate was seeded with appropriate cell densities of exponentially growing A549, HeLa, and HSC2 cells (5000–10000 cells/well). Stock test compounds (**1**–**5**) dissolved in dimethyl sulfoxide (DMSO) were screened at concentrations of 100, 10 and 1 µM after a 24 h incubation period at 37 °C with 5 percent CO_2_, followed by varying concentrations for the generation of concentration-dependent curves of the most cytotoxic compounds with culture medium (final DMSO concentration in medium = 0.1 percent, by volume). MTT solution in PBS (5 mg/mL) was added to each well after 48 h of incubation, and the incubation was continued for another 90 min. A phase contrast microscopic analysis verified the production of intracellular formazan crystals (mitochondrial reduction product of MTT). The medium was withdrawn at the conclusion of the incubation time, and 100 µL of DMSO were added to each well with shacking for 10 min to dissolve the produced formazan crystals (200 rpm). The absorbance at 492 nm (OD) of dissolved crystals was measured on a microplate reader (SunriseTM microplate reader, Tecan Austria Gmbh, Grödig, Austria) and utilized as a marker of cell proliferation.

#### 3.5.2. Anti-Proliferation Quantitative Analysis

IC_50_ values were obtained using GraphPad Prism^®^ v6.0 software and the concentration-response curve fit to the non-linear regression model (GraphPad Software Inc., San Diego, CA, USA).

## 4. Conclusions

Although there are few chemical studies that focus specifically on the species *S. convolutum*, the biological activity of key *Sarcophyton* metabolites has drawn extensive chemical analyses of the genus. Herein a solvent extract of the soft coral afforded seven cembrane-type diterpenoids, including sarcoconvolutum F and G (**1**,**2**), previously undescribed. Compound **4** exhibited modest cytotoxic activity with an IC_50_ of 56 μM and 55 µM against lung adenocarcinoma (A549) and oral cavity carcinoma (HSC-2) lines, respectively. 

## Figures and Tables

**Figure 1 molecules-27-05835-f001:**
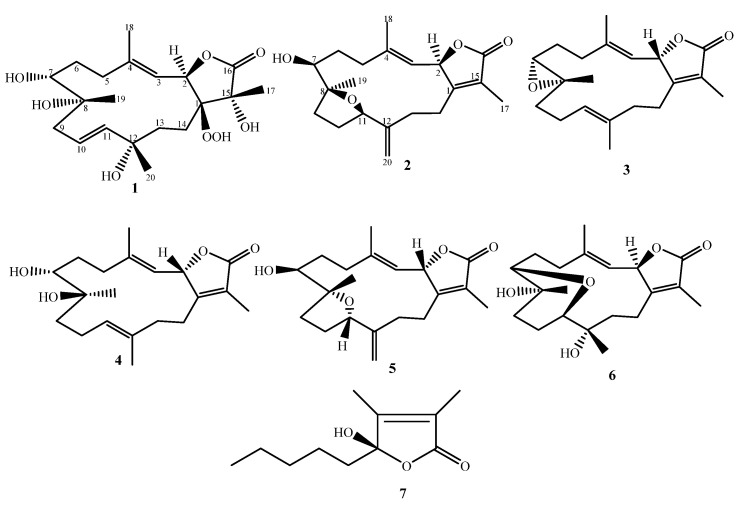
Structures of the isolates from *S. conovlutum*.

**Figure 2 molecules-27-05835-f002:**
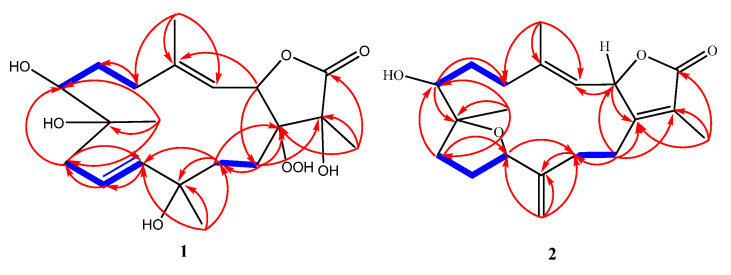
Observed ^1^H-^1^H COSY (bold blue line) and HMBC (red arrow) correlations for (**1**) and (**2**).

**Figure 3 molecules-27-05835-f003:**
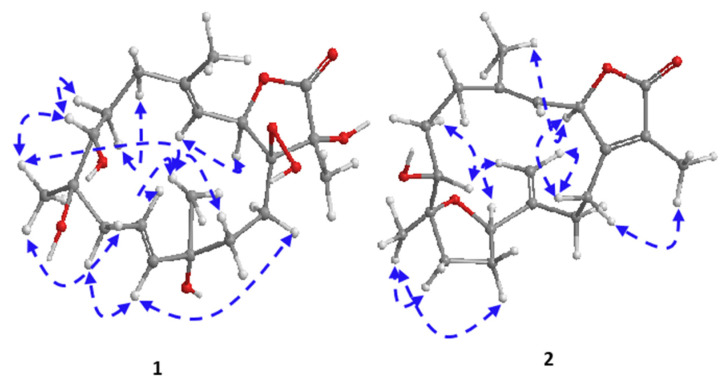
Significant NOESY correlations of (**1**) and (**2**).

**Figure 4 molecules-27-05835-f004:**
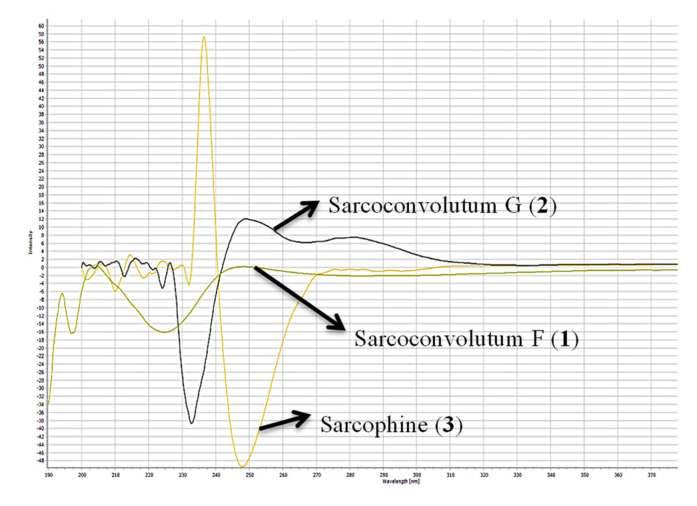
Circular dichroism (CD) spectra of (**1**–**3**).

**Figure 5 molecules-27-05835-f005:**
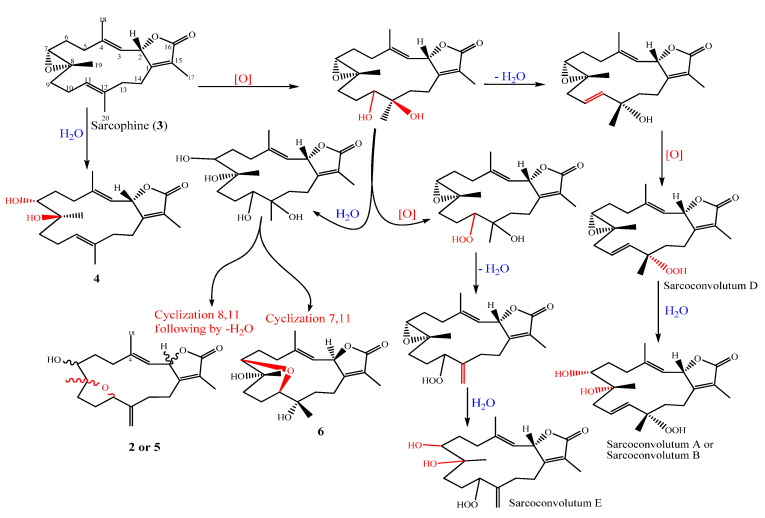
Putative biosynthetic pathways responsible for the synthesis of compounds (**2**–**6**) and other reported.

**Figure 6 molecules-27-05835-f006:**
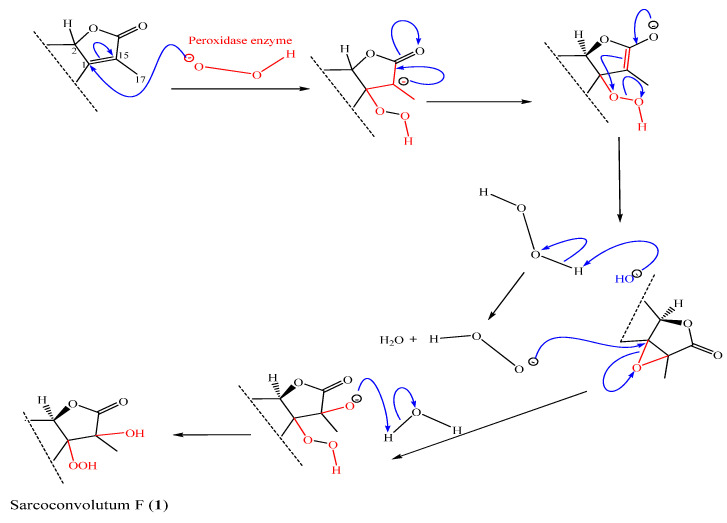
Biogenesis route of Sarcoconvolutum F (**1**).

**Table 1 molecules-27-05835-t001:** ^1^H and ^13^C NMR data of compounds **1** and **2** in CDCl_3_ (500 and 125 MHz *δ* in ppm, *J* in Hz).

No	1		2	
	*δ_H_*	*δ_C_*	*δ_H_*	*δ_C_*
1	---	86.9	------	162.0
2	5.04 br d (10.4)	80.9	5.47 d (9.8)	79.5
3	5.22 br d (10.4)	117.5	5.09 d (9.8)	120.4
4	---	143.5	------	145.9
5	2.17 m 2.42 ddd (13.3, 13.3, 3.7)	36.1	2.08 ddd (13.7, 13.7, 2.4) 2.44 ddd (13.7, 3.7, 3.7)	32.9
6	1.42 m 1.74 m	24.9	1.58 overlap 1.99 ddd (13.7,13.7,4.0)	33.4
7	3.64 br d (10.9)	69.7	3.22 brd (10.1)	74.6
8	---	73.9	-----	85.1
9	2.12 dd (14.5, 2.5) 2.52 dd (14.5, 9.1)	43.3	2.17 m, 1.72 m	36.1
10	5.64 dd (15.7, 3.7)	126.6	1.90 m 1.90 m	30.8
11	5.61 br d (15.7)	136.3	4.50 t (6.9)	83.1
12	---	79.9	----	148.6
13	1.92 ddd (17.2, 17.2, 3.7) 2.08 br d (17.2)	26.7	2.18 ddd (15.4, 13.1, 6.0) 2.26 ddd (15.4, 13.1, 4.0)	26.8
14	1.66 m 2.16 m	19.0	2.32 ddd (13.1, 13.1, 4.0) 2.66 ddd (13.1, 13.1, 6.0)	27.2
15	---	77.5	----	123.5
16	---	175.2	----	174.8
17	1.80 s	22.9	1.85 s	8.8
18	1.69 s	15.9	1.96 br s	20.7
19	1.21 s	22.7	1.16 s	19.8
20	1.16 s	27.3	4.88 br s 5.04 br s	111.1


## Data Availability

The data presented in this study are available in Appendix A.

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
