# Peer review of "Sarcoconvolutums F and G: Polyoxygenated Cembrane-Type Diterpenoids from Sarcophyton convolutum, a Red Sea Soft Coral"

_molecules, 2022, doi:10.3390/molecules27185835_

Round 1
Reviewer 1 Report
This manuscript is described about isolation and structure determination of sarcoconvolutums F and G from Sarcophyton convolutum. The authors carefully assigned all protons and carbons by measuring various NMR spectroscopies, determining structures finally as shown Figure 1. However, it seems a lack of evidence for structure determination. For example, the authors claim the configuration of H-2 is alpha. In this case, the reviewer considers that the NOE correlation between H-2 and H3-17 could be required, but the authors do not mention about the above correlation. In addition, it is difficult for the readers to understand why the authors claim H-13, 14, and 17 are a-orientation, because H-13 and 14 are geminal protons, respectively. Therefore, additional explanation for the above claims should be required. Moreover, putative biogenesis route described in figure 6 would be doubtful. It looks Sn2 reaction of tertiary-epoxide with peroxide, but Sn2 reaction on a tertiary-carbon does not occur. If it occurs, the relative stereochemistry of alcohol and peroxide should be anti. The reviewer considers this manuscript is required more evidence and explanation for both structure determination and proposed mechanism for publication in Molecules.
Author Response
To the Reviewer 1 #
Dear Prof. Thanks you for your help in advance.
Firstly, indeed we appreciate your points as will enhance our understanding for NMR spectroscopy.
I enclose here, the paragraph of your nice comments.
- This manuscript is described about isolation and structure determination of sarcoconvolutums F and G from Sarcophyton convolutum. The authors carefully assigned all protons and carbons by measuring various NMR spectroscopies, determining structures finally as shown Figure 1. However, it seems a lack of evidence for structure determination. For example, the authors claim the configuration of H-2 is alpha. In this case, the reviewer considers that the NOE correlation between H-2 and H3-17 could be required, but the authors do not mention about the above correlation.
Response 1:
The point of chirality and stereochemistry at C-2 rises from the CD spectrum comparison of 1 and Sarcophine (3) revealed the opposite absolute configuration for the two compounds at C-2, corroborating CD findings for 1 (Figure 4), [14, 22]. Carful investigation of the NOESY spectrum of 1, reflects that there is no correlation was observed between H-2a and the tertiary methyl at C-15. So, Both of H-2 and Me-17 should be in an anti-orientaion.
- In addition, it is difficult for the readers to understand why the authors claim H-13, 14, and 17 are a-orientation, because H-13 and 14 are geminal protons, respectively. Therefore, additional explanation for the above claims should be required.
Response 2: structure determination has been revised; According to your point we, reviesed the NOESY correlation with sarcoroseolide B [27]. Also we mention it's in the manuscript.
- Moreover, putative biogenesis route described in figure 6 would be doubtful. It looks Sn2 reaction of tertiary-epoxide with peroxide, but Sn2 reaction on a tertiary-carbon does not occur. If it occurs, the relative stereochemistry of alcohol and peroxide should be anti. The reviewer considers this manuscript is required more evidence and explanation for both structure determination and proposed mechanism for publication in Molecules.
Response 3: structure determination has been revised
Also, as you mention about the SN2 reaction mechanism (putative biogenesis) on the second step by the attack of nucelophile on the tertiary-epoxide in lactone ring may be happened. Perhaps, the adaption of the of opening the epoxy ring happen through protonated of it firstly, that causes the ring was opened followed by the attack of proxy anion. For the SN2 mechanism as you mention the orientation for both of the peroxy and hydroxy must be anti.(The stereochemistry at C-15 should be inverted). Especially, for the last point with the NOESY, we revised C-15 from the pro-S to be a pro-R. We find a document which may be support the idea
" SN2 Reactions at Tertiary Carbon Centers in Epoxides, Angew. Chem. Int. Ed., 2017, 56, 9719-9722.
Really, your comment help us that the biogenesis must be in fitting with NMR data along with the CD.
Thank you.
Reviewer 2 Report
This manuscript describes the isolation, structure elucidation, and cytotoxity bioassay of diterpenoids from the soft coral Sarcophyton convolutum. In this work, two new cembranoids together with five known analogues were discovered. Additionally, one compound exhibited cytotoxity activity towards A549 and HSC-2 cell lines. Despite of the grammar and typo errors and somewhat unreliable structure elucidation, this work is important, as it enriched the rapidly expanding library of marine diterpenoids. However, revisions are required if the article is considered for publication.
1. Title: It is better to use ‘Sarcoconvolutums F and G’
2. Abstract:
2.1. ‘…particularly for cancer and infectious disease pharmacotherapy's.’ ‘This order to support decision for…’ These were obscure.
2.2. Italic format for ‘Sarcophyton convolutum’
2.3. ‘sarcoconvolutum F (1) and G (2)’ → ‘sarcoconvolutums F (1) and G (2)’
2.4. ‘…isolated through using a variety of chromatographic techniques of the ethyl acetate extract of the soft coral Sarcophyton convolutum.’ → ‘…isolated from the ethyl acetate extract of the soft coral Sarcophyton convolutum using a variety of chromatographic techniques.’
2.5. ‘To estimate their structures, a combination of spectroscopic techniques were performed, including NMR, HREIMS, and CD, and/or NMR data was especially in comparison to previously published data.’ → ‘To determine their structures, a combination of spectroscopic techniques were performed, including NMR, HREIMS, and CD data, together with comparison with previously published data.’
2.6. ‘possess’ → ‘possessed’
2.7. It is better to give a brief summary of the results of the cytotoxic bioassay after the sentence ‘Furthermore, all the isolates (1-7) were screened against A549, HeLa and HSC-2 cancer cell lines’
3. Introduction:
3.1. This section is redundant, and some points were repeated. At the beginning of the first paragraph, the marine natural products of the Red Sea was introduced, but suddenly it turned to introduce the category of the corals, ending with summary of the secondary metabolites of the genus Sarcophyton. However, the secondary paragraph started as the historical study of cembrane-type diterpenes, without continuing from the above. The logical of this section was a bit confusing. Quite difficult to understand was the third paragraph. Please re-write the Introduction.
3.2. In addition, some revisions are listed as the following:
3.2.1. Italic format for Latin names ‘Sarcophyton’, ‘Sarcophyton glaucum’, ‘S. crassocaule’, 3.2.2. ‘Lobophytum’, ‘Sinularia’, ‘S. convolutum’.
3.2.3. ‘cembranediterpenes’ → ‘cembrane diterpenes’
3.2.4. ‘a Me groupor an exo-CH2= group’ → ‘a Me group or an exo-CH2= group’
3.2.5. ‘linkageintermediate’ → ‘linkage intermediate’
3.2.6. ‘cembranoidsbetween’ → ‘cembranoids between’
3.2.7.Numbers in subscript format: ‘CH2OH’, ‘CHCl3’
3.2.8. Please add a citation as the source reference of the compound sarcophyocrassolide A.
3.2.9. ‘plants’ → ‘soft corals’
3.2.10. ‘sarcoconvolutum F (1) & G (2)’ → ‘sarcoconvolutums F (1) and G (2)’
4. Structure Elucidation:
4.1. The structure elucidation of compound 1 was not logical. Why did the molecular formula reflect ‘a bicyclic frame skeleton’? Then without analysis of the NMR data, ‘Two double bonds are responsible for two of the four elements of unsaturation, giving rise in a bicyclic molecule.’ this conclusion was given. ‘The signal at δH 5.04 (d, J = 10.4 Hz, 1H) correlated with a proton signal at δH 5.22 (d, J = 10.4, 1H) and quaternary olephnic carbon at δC 143.5 (Fig. 2), respectively, allowed for the assignments of H-2, H-3, C-4.’ was obscure. To interpret the presence of the peroxide group in compound 1, ref. [43] was cited. However, no compound containing the peroxide group or closely resembled compound was reported in this reference. It is necessary to find new evidence to confirm it. In addition, compound 1 and the known compound 3 had different chromophores, and their CD character were also different as shown in Figure 4. Therefore, it is not reliable to determine the absolute configuration of compound 1 by comparison of the CD spectra. And for compound 2, it seemed improper to use the HREIMS data as evidence for the presence of the ether linkage between C-8 and C-11. Please re-write it.
4.2. Typo errors: ‘0C’, ‘[α]25D’.
4.3. ‘The carbon skeleton of all fractions would be the same, according to preliminary NMR investigation, with the degree of oxidation and the configuration of one or more chiral centres altering.’ This sentence was obscure.
4.4. The HRMS data of compound 1 described in the manuscript was not consistent with that shown in Figure S8.
4.5. The drawing of compound 6 in Figure 1 needed to be modified.
4.6. Please give the unit for the IC50 values, and revise the grammar errors.
4.7. The subtitle ‘2.1. Structure Elucidation of the Isolated Compounds’ could be deleted.
5. Please revise the references according to the Instructions for Authors.
Author Response
To the Reviewer 2 #
Thanks for respected reviewer for his comments to improve our manuscript.
This manuscript describes the isolation, structure elucidation, and cytotoxity bioassay of diterpenoids from the soft coral Sarcophyton convolutum. In this work, two new cembranoids together with five known analogues were discovered. Additionally, one compound exhibited cytotoxity activity towards A549 and HSC-2 cell lines. Despite of the grammar and typo errors and somewhat unreliable structure elucidation, this work is important, as it enriched the rapidly expanding library of marine diterpenoids. However, revisions are required if the article is considered for publication.
- Title: It is better to use ‘Sarcoconvolutums F and G’
Response: has been replaced to Sarcoconvolutums F and G
- Abstract:
2.1. ‘…particularly for cancer and infectious disease pharmacotherapy's.’ ‘This order to support decision for…’ These were obscure.
Response: Type a mistake was omitted.
2.2. Italic format for ‘Sarcophyton convolutum’
Response: Type a mistake have been corrected.
2.3. ‘sarcoconvolutum F (1) and G (2)’ → ‘sarcoconvolutums F (1) and G (2)’
Response: Type a mistake have been corrected
2.4. ‘…isolated through using a variety of chromatographic techniques of the ethyl acetate extract of the soft coral Sarcophyton convolutum.’ → ‘…isolated from the ethyl acetate extract of the soft coral Sarcophyton convolutum using a variety of chromatographic techniques.’
Response: the sentence have been revised.
2.5. ‘To estimate their structures, a combination of spectroscopic techniques were performed, including NMR, HREIMS, and CD, and/or NMR data was especially in comparison to previously published data.’ → ‘To determine their structures, a combination of spectroscopic techniques were performed, including NMR, HREIMS, and CD data, together with comparison with previously published data.’
Response: Type a mistake have been corrected
2.6. ‘possess’ → ‘possessed’
Response: Type a mistake have been corrected
2.7. It is better to give a brief summary of the results of the cytotoxic bioassay after the sentence ‘Furthermore, all the isolates (1-7) were screened against A549, HeLa and HSC-2 cancer cell lines’
Response: a brief summary of the results of the cytotoxic bioassay has been added.
- Introduction:
3.1. This section is redundant, and some points were repeated. At the beginning of the first paragraph, the marine natural products of the Red Sea was introduced, but suddenly it turned to introduce the category of the corals, ending with summary of the secondary metabolites of the genus Sarcophyton. However, the secondary paragraph started as the historical study of cembrane-type diterpenes, without continuing from the above. The logical of this section was a bit confusing. Quite difficult to understand was the third paragraph. Please re-write the Introduction.
Response: the Introduction has been modified according your comments and modified to fit the target of work. .
3.2. In addition, some revisions are listed as the following:
3.2.1. Italic format for Latin names ‘Sarcophyton’, ‘Sarcophyton glaucum’, ‘S. crassocaule’, 3.2.2. ‘Lobophytum’, ‘Sinularia’, ‘S. convolutum’.
Response: deleted during modified
3.2.3. ‘cembranediterpenes’ → ‘cembrane diterpenes’
Response: Type a mistake have been corrected
3.2.4. ‘a Me groupor an exo-CH2= group’ → ‘a Me group or an exo-CH2= group’
Response: Type a mistake have been corrected
3.2.5. ‘linkageintermediate’ → ‘linkage intermediate’
Response: deleted during modified
3.2.6. ‘cembranoidsbetween’ → ‘cembranoids between’
Response: Type a mistake have been corrected
3.2.7.Numbers in subscript format: ‘CH2OH’, ‘CHCl3’
Response: Type a mistake have been corrected
3.2.8. Please add a citation as the source reference of the compound sarcophyocrassolide A.
Response: Type a mistake have been corrected
3.2.9. ‘plants’ → ‘soft corals’
Response: deleted during modified
3.2.10. ‘sarcoconvolutum F (1) & G (2)’ → ‘sarcoconvolutums F (1) and G (2)’
Response: Type a mistake have been corrected
- Structure Elucidation:
4.1. The structure elucidation of compound 1 was not logical. Why did the molecular formula reflect ‘a bicyclic frame skeleton’? Then without analysis of the NMR data, ‘Two double bonds are responsible for two of the four elements of unsaturation, giving rise in a bicyclic molecule.’ this conclusion was given.
Response: The correct molecular formula was re-evaluated according to the HREIMS and the value faced the attachment supplementary data. Also, we rewrite it and discussed in the manuscript.
‘The signal at δH 5.04 (d, J = 10.4 Hz, 1H) correlated with a proton signal at δH 5.22 (d, J = 10.4, 1H) and quaternary olephnic carbon at δC 143.5 (Fig. 2), respectively, allowed for the assignments of H-2, H-3, C-4.’ was obscure.
Response : structure determination has been revised; According to your point we, revised the NOESY correlation with sarcoroseolide B [27]. Also we mention it's in the manuscript.
To interpret the presence of the peroxide group in compound 1, ref. [43] was cited. However, no compound containing the peroxide group or closely resembled compound was reported in this reference. It is necessary to find new evidence to confirm it.
Response : structure determination has been revised; According to your point we, revised the skeleton according to the compound given a trivial name sarcoroseolide B [27]. Also we mention it's in the manuscript with the delete of ref. 43, no need to mention it in the manuscript.
In addition, compound 1 and the known compound 3 had different chromophores, and their CD character were also different as shown in Figure 4. Therefore, it is not reliable to determine the absolute configuration of compound 1 by comparison of the CD spectra.
The point of chirality and stereochemistry at C-2 rises from the CD spectrum comparison of 1 and Sarcophine (3) revealed the opposite absolute configuration for the two compounds at C-2, corroborating CD findings for 1 (Figure 4), [14, 22]. Carful investigation of the NOESY spectrum of 1, reflects that there is no correlation was observed between H-2a and the tertiary methyl at C-15. So, Both of H-2 and Me-17 should be in an anti-orientaion.
And for compound 2, it seemed improper to use the HREIMS data as evidence for the presence of the ether linkage between C-8 and C-11.
Response: The presence of an ether-linkage was determined through the comparison of NMR data of a previously reported by us crassumols G [29].
Please re-write it.
4.2. Typo errors: ‘0C’, ‘[α]25D’.
Response: deleted during modified
4.3. ‘The carbon skeleton of all fractions would be the same, according to preliminary NMR investigation, with the degree of oxidation and the configuration of one or more chiral centres altering.’ This sentence was obscure.
Response: Type a mistake the sentence has been modified.
4.4. The HRMS data of compound 1 described in the manuscript was not consistent with that shown in Figure S8.
Response: Type a mistake have been corrected
4.5. The drawing of compound 6 in Figure 1 needed to be modified.
Response: Type a mistake the drawing of compound 6 in Figure 1 has been modified.
4.6. Please give the unit for the IC50 values, and revise the grammar errors.
Response: Type a mistake the grammar has been revised.
4.7. The subtitle ‘2.1. Structure Elucidation of the Isolated Compounds’ could be deleted.
Response: The subtitle has been deleted.
- Please revise the references according to the Instructions for Authors
Response: The References has been Revised.
Round 2
Reviewer 1 Report
The revised manuscript includes additional explanations for structure determination, therefore, the reviewer suggests acceptance of this manuscript for publication.
However, it is still difficult for the reviewer to understand the Sn2 reaction of epoxide in the putative biosynthesis. Of course, the reviewer also agrees with the mechanism shown in reference (Angew), but, the conditions are entirely different from putative biosynthesis. For example, the enolate could also be oxidized to form an epoxide, leading to an alpha-hydroxy ketone without via the epoxide the authors mentioned.
Author Response
Reviewer 1
Dear Prof. Thanks for your help in advance.
The revised manuscript includes additional explanations for structure determination, therefore, the reviewer suggests acceptance of this manuscript for publication.
However, it is still difficult for the reviewer to understand the Sn2 reaction of epoxide in the putative biosynthesis. Of course, the reviewer also agrees with the mechanism shown in reference (Angew), but, the conditions are entirely different from putative biosynthesis. For example, the enolate could also be oxidized to form an epoxide, leading to an alpha-hydroxy ketone without via the epoxide the authors mentioned.
Response: The comment takes into our best consideration. your valuable comment and your suggestion was invloved in the mauscript to enhance the configuration skeleton.
[An alternative biosynthetic route would be that the enolate is oxidized directly to form an epoxide, leading to an alpha-hydroxy ketone without via an epoxide intermediate (not shown).]
We appreciate your help in advance
Reviewer 2 Report
Although the resubmitted manuscript has been corrected/improved by the authors to some extent, there are still a certain number of comments to be addressed by the authors before the manuscript is considered for publication.
From the authors’ responses, the concerns are listed here.
1. What were the names for compounds 3–6. The authors should give a brief introduction in the main text.
2. Why was the hydroxyl group at C-7 α-orientated and the peroxy group at C-1 β-orientated? Please give detailed interpretations, especially for the peroxy group.
3. What were the CD findings for 1? The CD Cotton effects depends on the chromophores. The reference compound sarcophine (3) possessed an α,β-unsaturated butenolactone moiety, and displayed one Cotton effect at 248 nm and the other one at 235 nm as shown in Figure 4. However, compound 1 displayed only one Cotton effect at 225 nm.
4. Please add the unit ‘Hz’ after the coupling constant(s), such as ‘…J = 10.4…’and ‘…J = 10.4…’ on P5. Please check it throughout the whole manuscript.
5. Please revise ‘…H3-17 and carbon signals at δC 86.9)…’ as ‘…H3-17 and carbon signals at C-1 (δC 86.9)…’. Please check it throughout the whole manuscript, such as ‘…the methyl protons H-19 (δH 1.21)…’ and ‘…methyl protons H-20 (δH 1.16)…’.
6. ‘The presence of an ether linkage was suggested by a high downfield carbon signals at δC 83.1 and 85.1, which were functionally validated by HREIMS’. As this sentence said, the reference compound should be added herein.
7. ‘A cis configuration between the γ-lactone (H-2) and the olefinic proton (H-3) was established using a vicinal coupling constant of 9.8 Hz between H-2 (δH 5.47) and H-3 (δH 5.09) as well as a NOESY correlation of α-orientation of H-2 with H3-18 (δH 1.96).’ However, H-3 and H-2 were on opposite orientations as shown in Figure 3.
8. ‘10.1 Hz vicinal coupling constant between H-7 (δH 3.22) and H-6 (δH 1.58), as well as 13.7 Hz vicinal coupling between H-6 (δH 1.99) and H-5 (δH 2.08).’ What did this sentence mean?
9. Please add the unit in the sentence ‘…IC50 values of 56 and 55…’ on P7.
10. Please don’t forget to revise the styles of references according to the Instructions for Authors.
11. Please revise the title in the document Supplementary Materials, and the words ‘1H 1H COSY’ in figure legends.
Author Response
Reviewer 2
Thanks for respected reviewer for his comments to improve our manuscript.
Although the resubmitted manuscript has been corrected/improved by the authors to some extent, there are still a certain number of comments to be addressed by the authors before the manuscript is considered for publication.
From the authors’ responses, the concerns are listed here.
- What were the names for compounds 3–6. The authors should give a brief introduction in the main text.
Response: The trivial names of compounds 3-7 was added in the manuscript with their references.
- Why was the hydroxyl group at C-7 α-orientated and the peroxy group at C-1 β-orientated? Please give detailed interpretations, especially for the peroxy group.
Response: Really, the gross structure of the skeleton configuration was established by:
- There was no correlation between H-2 and Me-17 in the NOESY spectrum of compound 1, so, we expect that the orientation should be opposite to each other's.
- Comparison of the CD spectrum of compound 1 with sarcophine (3), resulted in the orientation of H-2 should be in alpha position and depending on the NOESY spectrum of 1, Me-17 should be in beta orientation.
- The putative biosynthesis of 1 following the route of SN2 reaction of epoxide, suggested that A Walden inversion occurs at a tetrahedral carbon atom during an SN2 By this idea both of hydroxyl and peroxy groups should be anti-orienation. (Fig. 6).
- In focus, there was a close deep similarity, especially the 13C NMR data of compound 1 with sarcoroseolide B was isolated previously from roseum [31], with the exception that 1 missing the oxygen bridge which was found in sarcoroseolide B expected to be replaced by the peroxy group at C-1. Moreover, compound 1 possessed an 34 amu in its molecular weight increases than sarcoroseolide B, which predict that 1 having a peroxy group. Besides, functionalities with a molecular formula of sarcoroseolide B suggesting a tricyclic structure compared with 1 validated by HREIMS to be a bicyclic frame skelton.
- It seems that compound 1 precursory a resulted from the enzymatically hydrolysis from sarcoroseolide A was isolated previously from roseum [31].
- What were the CD findings for 1? The CD Cotton effects depends on the chromophores. The reference compound sarcophine (3) possessed an α,β-unsaturated butenolactone moiety, and displayed one Cotton effect at 248 nm and the other one at 235 nm as shown in Figure 4. However, compound 1displayed only one Cotton effect at 225 nm.
Response: yeah, the comment is totally in target,
For sarcophine (3) possessed an α,β-unsaturated butenolactone moiety, displayed a negative cotton effect at 248 response for the double bond and the other positive cotton effect at 235 related for C-2. As compound 1 displayed only one a negative cotton effect at 225 nm response for C-2. Based on the comparison of compound 1 and sarcophine (3) at nearly 225 nm resuled in the configuration at C-2 of it should be opposite.
- Please add the unit ‘Hz’ after the coupling constant(s), such as ‘…J = 10.4…’and ‘…J = 10.4…’ on P5. Please check it throughout the whole manuscript.
Response: Type a mistake have been corrected
- Please revise ‘…H3-17 and carbon signals at δC86.9)…’ as ‘…H3-17 and carbon signals at C-1 (δC 86.9)…’. Please check it throughout the whole manuscript, such as ‘…the methyl protons H-19 (δH 1.21)…’ and ‘…methyl protons H-20 (δH 1.16)…’.
Response: Type a mistake have been corrected in the whole manuscript.
- ‘The presence of an ether linkage was suggested by a high downfield carbon signals at δC83.1 and 85.1, which were functionally validated by HREIMS’. As this sentence said, the reference compound should be added herein.
Response: the reference has been added.
- ‘A cis configuration between the γ-lactone (H-2) and the olefinic proton (H-3) was established using a vicinal coupling constant of 9.8 Hz between H-2 (δH5.47) and H-3 (δH5.09) as well as a NOESY correlation of α-orientation of H-2 with H3-18 (δH 1.96).’ However, H-3 and H-2 were on opposite orientations as shown in Figure 3.
Response: the correlation between H-2 and H3-18 has been revised to H-2 and H-3 (cis configuration).
- ‘10.1 Hz vicinal coupling constant between H-7 (δH3.22) and H-6 (δH1.58), as well as 13.7 Hz vicinal coupling between H-6 (δH 1.99) and H-5 (δH 2.08).’ What did this sentence mean?
Response: the sentence was revised in the manuscript to be as follow” The value of vicinal coupling constant which equally to 10.1 Hz between H-7 (δH 3.22) and H-6 (δH 1.58), as well as 13.7 Hz vicinal coupling between H-6 (δH 1.99) and H-5 (δH 2.08) with the aid of NOESY spectrum supported the orientation of H-7 in a-orientation and differentiate between C-6 protons chemical shifts together with the comparison with those of crassumol G [24].
- Please add the unit in the sentence ‘…IC50values of 56 and 55…’ on P7.
Response: Type a mistake have been added.
- Please don’t forget to revise the styles of references according to the Instructions for Authors.
Response: references were done by the End-Note style of MDPI- molecules
- Please revise the title in the document Supplementary Materials, and the words ‘1H 1H COSY’ in figure legends.
Response: the title in the document Supplementary Materials, and the words ‘1H 1H COSY’ in figure legends have been revised.
Round 3
Reviewer 2 Report
The resubmitted manuscript has been improved. Only minor revisions are required.
1. P2: Revise ‘Sarcophine’ and ‘Crassumol G’ as ‘sarcophine’ and ‘crassumol G’.
2. Remove the number ‘1’ on the left half of Figure 3.
3. Rewrite the sentence ‘which is a common cembrane diterpenoids found in many soft coral diverse Sarcophyton genus collecting from different regions’.
4. Revise ‘Figures S1–S27: NMR spectra for compounds 1-7’ as ‘Figures S1–S27: NMR spectra for compounds 1-7 and MS spectra for compounds 1 and 2’
5. Revise ‘1H–1H–COSY’ as ‘1H–1H COSY’ in the document Supplementary Materials.
Author Response
Thanks for respected reviewer for his comments to improve our manuscript.
- P2: Revise ‘Sarcophine’ and ‘Crassumol G’ as ‘sarcophine’ and ‘crassumol G’.
Response 1: Type a mistake have been corrected
- Remove the number ‘1’ on the left half of Figure 3.
Response 2: Type a mistake have been removed
- Rewrite the sentence ‘which is a common cembrane diterpenoids found in many soft coral diverse Sarcophyton genus collecting from different regions’.
Response 3: the sentence was revised in the manuscript to be as follow” Which is a typical cembrane diterpenoids discovered in many different types of soft coral belonging to thr Sarcophyton genus and accumulating from various locations"
- Revise ‘Figures S1–S27: NMR spectra for compounds 1-7’ as ‘Figures S1–S27: NMR spectra for compounds 1-7and MS spectra for compounds 1 and 2’
Response 4: the sentence was revised in the manuscript to be as follow ‘Figures S1–S27: NMR spectra for compounds 1-7 and MS spectra for compounds 1 and 2’
- Revise ‘1H–1H–COSY’ as ‘1H–1H COSY’ in the document Supplementary Materials.
Response 5: Type a mistake have been revised in the document Supplementary Materials.